

# Impact of asymmetric uncertainties in ice sheet dynamics on regional sea level projections

Renske de Winter[1], Thomas J. Reerink[2], Aimée B.A. Slangen[3], Hylke de Vries[4], Tamsin Edwards[5], and Roderik S.W. van de Wal[2]

[1]Institute for Marine and Atmospheric research Utrecht, Physical Geography, Utrecht University, Heidelberglaan 2, 3584 CS Utrecht, The Netherlands

[2]Institute for Marine and Atmospheric research Utrecht, Utrecht University, Princetonplein 5, 3584 CC Utrecht, The Netherlands

[3]Royal Netherlands Institute for Sea Research (NIOZ), Department of Estuarine & Delta Systems (EDS), Yerseke, The Netherlands and Utrecht University, The Netherlands

[4]Royal Netherlands Meteorological Institute (KNMI), P.O. Box 201, 3730 AE De Bilt, The Netherlands

[5]Open University, Milton Keynes, UK

*Correspondence to:* Renske de Winter (r.c.dewinter@uu.nl)

**Abstract.** Currently a paradigm shift is made from global averaged to spatially variable sea level change (SLC) projections. Traditionally, the contribution from ice sheet mass loss to SLC is considered to be symmetrically distributed. However, several assessments suggest that the probability distribution of dynamical ice sheet mass loss is asymmetrically distributed towards higher SLC values. Here we show how asymmetric probability distributions of dynamical ice sheet mass loss impact the high-end uncertainties of regional SLC projections across the globe. For this purpose we use distributions of dynamical ice sheet mass loss presented by Church et al. (2013), De Vries and Van de Wal (2015) and Ritz et al. (2015). The global average median can be 0.18m higher compared to symmetric distributions based on IPCC-AR5, however the change in the global average $95^{th}$ percentile SLC is considerably larger with a shift of 0.32m. Locally the $90^{th}$, $95^{th}$ and $97.5^{th}$ SLC percentiles exceed +1.4, +1.6 and +1.8m. The high-end percentiles of SLC projections are highly sensitive to the precise shape of the probability



distributions of dynamical ice sheet mass loss. The shift towards higher values is of importance for coastal safety strategies as they are based on the high-end percentiles of projections.

# 1 Introduction

The impacts of sea level rise (SLC) will be one of the major impacts of climate change in the $21^{th}$ century (Nicholls et al.,
2011; Cazenave and Le Cozannet, 2014). Coastal safety standards are often formulated by analysing the high percentiles of the probability distribution, resulting in magnitudes of events with an acceptable return frequency (Katsman et al., 2011). These types of studies are executed in order to analyse events that are infrequent, but are expected to have a large impact on economy and society (Jonkman et al., 2011). Including high-end SLC projections is therefore the logical next step in coastal safety analysis. This requires two aspects; the transformation from global average SLC projections to regional SLC projections and
providing insight of the uncertainties of these regional SLC.

Local impact studies (e.g. de Winter and Ruessink, 2017) show that the amount of SLC may affect coasts and the corresponding mitigation measures significantly. This emphasizes the need for regional SLC projections, since the amount of SLC can deviate from global average values due to changes in ocean currents, thermal expansion, and gravitational and rotational effects induced by land ice and terrestrial ground and surface water mass changes (Mitrovica et al., 2001). Recent studies show
those spatially variable SLC projections (Slangen et al., 2012; Perrette et al., 2013; Slangen et al., 2014; Lyu et al., 2014; Kopp et al., 2014; Grinsted et al., 2015).

There are several components that contribute to SLC: surface mass balance changes of glaciers and ice sheets, global steric plus dynamic topography and atmospheric pressure, groundwater depletion, glacial isostatic adjustment (GIA) and dynamical ice sheet mass loss. Particularly the uncertainty of the last component, dynamical ice sheet mass loss, is under debate. Tra-
ditionally, the contribution from dynamical ice sheet mass loss to SLC is assessed by analysing the median and the standard deviation. Two independent expert-judgment based studies (Bamber and Aspinall, 2013; Horton et al., 2014) and a model based assessment (Ritz et al., 2015) concluded that the probability distributions of the ice dynamical contribution may be asymmetrical. The method in the expert-judgment based studies is criticised (Gregory et al., 2014; Clark et al., 2015; De Vries and Van de Wal, 2015), but from a physical point of view it cannot be excluded that the ice dynamical contribution of ice sheets
has a larger uncertainty towards higher-SLC values (Jacobs et al., 2011; Ritz et al., 2015; Pollard et al., 2015) than considered so far. At the same time there is a large difference between the expert-informed dynamical ice sheet mass loss and most of the numerical modelling studies (Little et al., 2013; Golledge et al., 2015; Ritz et al., 2015). Particular, DeConto and Pollard (2016) project high values due to ice cliff instability in combination with parameterizations for rapid ice shelf disintegration. These higher projections are primarily caused by the possible collapse of marine-based sectors of the Antarctic ice sheet (AIS)
(Church et al., 2013; Favier et al., 2014). An asymmetric probability distribution for the Greenland ice sheet (GIS) can also not be excluded (Nick et al., 2013). This implies that there is a greater uncertainty for events with an uncertainty above one standard deviation in the future contribution of dynamical ice mass changes to SLC than previously assumed, which will influence the projections of higher percentiles of the SLC probability distribution (Bamber and Aspinall, 2013; Horton et al., 2014). As



such, it is necessary to examine the consequences of asymmetric probability distributions for the ice dynamical contribution
on regional sea level projections.

Higher percentiles of the probability distribution are used to study uncertainties of SLC-projections, in line with coastal
safety assessments that use a return-frequency based approaches to determine safety levels (Nicholls et al., 2011; Hinkel et al.,
2015). Previous studies of the high-end percentiles of SLC projections for specific locations like the Netherlands (de Vries
et al., 2014) or Northern Europe (Grinsted et al., 2015) and at a network of tide-gauge sites (Kopp et al., 2014), show that
adopting asymmetric distributions for ice sheet mass loss have large impacts on high percentiles of SLC projections. Kopp
et al. (2014) use a data assimilation technique for tide-gauge sites combining historical data of sea levels with IPCC-AR5
SLC projections and expert judgement analysis of Bamber and Aspinall (2013) to estimate the impact of the ice dynamical
contribution. They use the study of Bamber and Aspinall (2013) to calibrate the shape of tail of the distribution and concluded
that at most location uncertainties of future SLC projections are driven by uncertainties in the ice sheet contribution. In contrast
to Kopp et al. (2014), Grinsted et al. (2015) use distribution presented by Bamber and Aspinall (2013), to project regional SLC
pattern of Northern Europe and the uncertainty ranges therein. They concluded that with the distribution of by Bamber and
Aspinall (2013), the $95^{th}$ percentile may be an additional 0.9 m above median. Both studies assume the contribution of the AIS
to be scenario independent.

Here we rely on a re-interpretation of the Bamber and Aspinall (2013) data as presented by De Vries and Van de Wal
(2015) and the data by Ritz et al. (2015). The main objective of this paper is to analyse the sensitivity of higher percentile of
regional SLC projections to asymmetric probability distributions for dynamical ice sheet mass loss by comparing probability
distributions of Church et al. (2013), De Vries and Van de Wal (2015) and Ritz et al. (2015). Especially the development of the
tail of the probability distribution is of interest, since these higher percentiles are, in contrast to the mean or median, often used
to determine safety standards.

## 2 Methods

### 2.1 Components contributing to sea level rise

In order to make a comparison between symmetric and asymmetric contributions of dynamical ice sheet mass loss to SLC all
other components contributing to SLC are kept the same for all simulations. Regional SLC fields of Slangen et al. (2014)(their
Fig 1) under RCP8.5 (Representative Concentration Pathway) climate scenario (Moss et al., 2010) are used for all contributions
except dynamic ice sheet mass loss. The regional SLC fields of Slangen et al. (2014) include contributions to SLC from:
surface mass balance of glaciers and ice sheets under RCP8.5 (their Fig. 1b and 2b); global steric plus dynamic topography and
atmospheric pressure loading under RCP8.5 (their Fig. 1d and 2d); scenario-independent groundwater depletion (their Fig. 1f
and 2f) and scenario-independent Glacial Isostatic Adjustment (GIA)(their Fig. 1g and 2g).

The normal, symmetric contributions for dynamical ice sheet mass loss are based on the median and likely range from
IPCC-AR5 (Church et al., 2013, their Table 13.5) (green, dash-dotted line Fig. 1a-b and e-f). Over the last few years, several
new probability density functions (PDFs) of the contribution of dynamical ice sheet mass loss to SLC are published (Bamber





and Aspinall, 2013; De Vries and Van de Wal, 2015; Ritz et al., 2015). These PDFs have a shift in median and asymmetry (e.g. Bamber and Aspinall (2013); De Vries and Van de Wal (2015); Ritz et al. (2015) Fig. 1). The PDFs show the skewness / asymmetry of a distribution, whereas changes in higher percentiles (the right tail of the PDF) are visible in the cumulative density function (CDF) of Fig. 1. The PDFs of De Vries and Van de Wal (2015) (hereafter VW15) are chosen to study impacts

of an asymmetric contribution of ice sheet mass loss on higher percentiles of SLC projections, because this data set contains distributions of all ice sheets (WAIS, EAIS and GIS). VW15 reanalysed the data from Bamber and Aspinall (2013) (hereafter BA13). As the expert judgments in BA13 were not weighted equally, VW15 more rigorously estimated the lack of consensus in the projection by fitting a log-normal distribution to the data and deriving uncertainties for the different levels of confidence. The effect of different input distributions on high-end SLC percentiles is analysed by comparing the SLC projection composed

with a dynamical ice sheet contribution of WAIS, EAIS and GIS according to VW15 and with the SLC projections containing probability distributions of Ritz et al. (2015) for WAIS and EIAS and the GIS contribution of IPCC-AR5.

Mass loss of an ice sheet does not result in a globally-uniform rise in sea level as a result of the gravitational effect, the added water mass will be redistributed according to a geographical pattern, the so-called fingerprint. Fingerprints of each ice sheet (Slangen et al., 2014) are used to convert the global projections of ice sheet mass loss to regional sea-level projections.

IPCC-AR5 (Church et al., 2013) does not make a distinction between West and East Antarctica (WAIS and EAIS) for the ice dynamical contribution. The contribution to SLC is assumed to originate from WAIS, as relative to the EAIS, this ice sheet is generally considered to make a larger contribution to SLC.

The regional SLC fields of Slangen et al. (2014) and the symmetric IPCC contributions are projections for the period $1990-2010$ to $2080-2100$. The asymmetric distributions of VW15 and Ritz et al. (2015) contain sea level changes in mm per

year in 2100. In order to be consistent with the projections of Church et al. (2013) andSlangen et al. (2014), a linear increase in SLC-rates between 2010 and 2100 is assumed, so changes between 2010 and 2090 due to dynamical ice sheet mass loss could be analysed.

Two aspects influence changes in the median between the PDFs constructed with symmetric and asymmetric components. First of all, the medians of the asymmetric projections for ice sheet mass loss are higher compared to the symmetric IPCC

distributions (Figure 1e-h). Secondly, even if the medians of the input PDFs are the same, the final PDF for the SLC projections might be different as a result of the asymmetry yielding higher SLC projections for higher percentiles.

Finally it is important to note that, we first assume that all components of SLC are uncorrelated, eventually a correlation between climate driven projections of SLC and ice dynamical contributions of SLC is also investigated (Section 3.3).

## 2.2 SEAWISE: Combining probability distributions

Future probability distributions of regional SLC are calculated by combining the probability distributions of the different components that contribute to sea level changes. For this analysis the SEAWISE model is developed. Computations are done on a global grid, with a grid size of $1°$ in longitude and latitude. The combined distribution $P_{com}(x)$ consists for each $x$ of all possible contributions of two independent distributions $P_1(x_1)$ and $P_2(x_2)$ for which the summed $x$-axis values, $x_1$ and $x_2$ add





to $x$. $P_{com}(x)$ is determined by looping over all combinations $x = x_1 + x_2$:

$$P_{com}(x) = \sum_{m=c}^{d} P_1(x - m\Delta x) P_2(m\Delta x) \tag{1}$$

where $x$ is the sea level change and $P_1$ and $P_2$ are the probability distributions for SLC of two individual components. See for example Figure 2, where the PDF of panel a and b are combined to calculate the PDF in panel c. Selecting 99.9% of the integrated distribution around the mode (the peak of the PDF) defines for each probability distribution a left and right boundary. The distribution is normalised on this selected interval. The left and right boundaries of $P_1$ are defined as $x_a$ and $x_b$, for $P_2$ the left and right boundaries are defined as $x_c$ and $x_d$. The interval counter, $m$ runs between $c$ and $d$ corresponding with $x_c$ and $x_d$, while taking $x_2 = m\Delta x$ and $x_1 = x - m\Delta x$. To obtain the entire distribution $P_{com}(x)$ Eq.1 is calculated from $x_a$ up to $x_b + x_d$.

This combined SLC probability distribution can be combined with a third SLC probability (e.g. Fig. 2d) distribution by a recursive approach, and so on, until all components that contribute to SLC are combined and $P_{total}$ is created (e.g. Fig. 2g). The supplementary material provides more details on how SEAWISE combines probability distributions if the components are assumed to be correlated.

For the regional projections, as depicted in Fig. 3 $P_{total}$ is saved for all SLC-values. For the global projection, per grid point $P_{total}$ is determined to calculate the specified percentiles (e.g. $90^{th}$).

## 3 Results

### 3.1 Changes in median

Combining the SLC probability distribution for the symmetric ice sheet contribution with the probability distribution for all other components to SLC results in a area-averaged global median ($50^{th}$ percentile) SLC of +0.68m in 2090 (Fig 4a). This is slightly higher than the +0.63m projected in IPCC-AR5 (their Table 13.5 Church et al., 2013). The difference between these projections results from larger projected SLC contributions from glaciers and groundwater depletion in Slangen et al. (2014). Most of the regional SLC projections have a higher median for the simulations where the contribution of ice sheet mass loss to SLC is considered to be asymmetric according to VW15. Using the asymmetric VW15 data rather than the symmetrical IPCC-based distributions for the ice dynamical components results in an area-average global median of +0.86m (Fig 4b). This shift in median of 0.18 m (Fig 4c) indicates that the estimate we use for the contribution of ice sheet mass loss is directed to higher values when using the asymmetric components. The explanation for this is twofold. First of all, medians of the contribution following the ice dynamical mass loss of IPCC-AR5 are 0.70 mm/yr and 0.82 mm/yr for GIS and AIS, respectively. In contrast to median contributions of 2.39 mm/yr for GIS, 1.49 mm/yr for EAIS and 0.17 mm/yr for WAIS for the VW15 distributions (Fig. 1e-h). Secondly, even if the medians are assumed to be the same, the asymmetry towards higher SLC values results in a small shift of the median in the combined projection.



More important than the average shift in median is that both regional SLC projections (with symmetric and asymmetric components) show large regional variability. These spatial variations (Fig. 4a-b) are the result of changes in: global steric plus dynamic topography and atmospheric pressure loading, surface mass balance of glaciers and ice sheets, groundwater depletion, Glacial Isostatic Adjustment (Slangen et al., 2014) and the impact of the ice dynamical contribution. The latter is because mass redistribution from the land to the ocean does not result in a globally uniform increase in sea level (Slangen et al., 2012; Perrette et al., 2013; Slangen et al., 2014; Lyu et al., 2014). Near an ice sheet for instance, mass loss of this ice sheet will result in a sea level fall, since the gravitational pull of the ice sheet becomes less when the mass decreases. In the far field of an ice sheet the rise of sea level will be above average. Using symmetric IPCC-based dynamical ice sheet mass loss contribution to SLC, the median SLC varies regionally from -1.07 to +1.03m (Fig. 4a). Assuming the VW15 asymmetric dynamical ice sheet mass loss, the median SLC projection ranges regionally from -1.90 m to +1.03m (Fig. 4b), with the largest differences in the Central Pacific and in the Artic Ocean.

## 3.2 Changes in higher percentiles

Changes in the tails of the probability distribution are much larger than the shift in median, as indicated by the probability distribution for 3 locations (Fig. 3). At the Denmark Strait (North West of Iceland), the GIS contribution to SLC is negative, the gravitational pull of GIS becomes less when mass is lost, subsequently less water is attracted towards GIS and sea levels are projected to fall compared to present day, at this location. On the other hand the EAIS and WAIS have a strong positive contribution at this location(Fig. 2), as the Denmark Strait is in the far field of EAIS and WAIS. As a result the GIS-contribution has a long tail towards negative values (Fig. 2a and Fig. A1b-c), whereas EAIS and WAIS contribute positively to SLC (Fig. A1d-e). Consequently, the total probability distribution including the asymmetric VW15 contribution of dynamical ice sheet mass loss to SLC is broader compared to the probability densities that include IPCC-AR5 values for dynamic ice sheet mass loss (Fig. 3b).

At locations where the GIS contribution is near zero (e.g., New York Bight, Fig. 3c), the shape of the tail in the combined distribution of the asymmetric simulations is dominated by the AIS contribution (Fig. SA1e). The largest changes occur where the contributions of the GIS, EAIS and WAIS are all positive (Fig. SA1). At these locations, for example the East Pacific ocean, the tail of the probability distribution becomes skewed towards higher SLC values (Fig. 3d).

The spatial pattern of the (change in) the higher percentiles including the asymmetric VW15 dynamical ice sheet contribution are shown in Fig. 5. The $90^{th}$, $95^{th}$ and $97.5^{th}$ percentiles of SLC locally exceed +1.4, +1.6 and +1.8m in large parts of the ocean (Fig. 5a-c). Asymmetric VW15 distributions of ice sheet mass loss alter the global average $90^{th}$, $95^{th}$ and $97.5^{th}$ percentiles by +0.27, +0.32 and +0.39m respectively, compared to the symmetric, IPCC-AR5 probability distributions (Fig. 5d-f). This is considerably larger than the shift in the median, which has a global average of +0.18m. It also shows that the difference increases for larger percentiles. The asymmetric VW15 distribution results in a shift of the upper percentiles to lower values near GIS and WAIS. This is the result of a decrease in the gravitational attraction due to the mass loss of the neighbouring ice sheet and hence a negative contribution to SLC. The change in higher percentiles is however positive in other places, with a maximum where the fingerprints of the different ice sheet are all larger than one. For the asymmetric VW15



projections the increase between the $90^{th}$ and $95^{th}$ percentile is of the same magnitude as the increase between the $95^{th}$ and $97.5^{th}$ percentile.

Adopting an asymmetric compared to a symmetric ice dynamical contribution increases the median by a globally-average value of +0.18m Fig. 4c. However, this difference is much smaller than the difference in the higher percentiles, with a shift of
the globally-average $97.5^{th}$ percentile of +0.39m.

We corrected the change of the higher percentiles for the change in local median SLC for each location, see Fig. 6 where Figure 4c is subtracted from Figure 5a-c. Generally, the corrected higher percentiles of SLC are still much larger compared to the projections with symmetric IPCC-based ice dynamical contributions, with changes up to 0.3m (Figure 6d-f). For both analysis (corrected and uncorrected for changes in the median), the increase to higher SLC values becomes more pronounced
for higher percentiles. This has large implications for the high-end uncertainties of future SLC projections, since with a higher mean sea level the allowance for extreme events become lower as critical thresholds are exceeded more frequently.

### 3.3   Correlation between ice sheet mass losses and steric SLC

Based on the IPCC-AR5 (Church et al., 2013) we initially assume the dynamical ice sheet contribution to SLC to be independent of climate change induced changes in SLC. Recent studies (Bamber and Aspinall, 2013), however, suggest a dependency
between climatological changes and dynamical ice sheet mass loss as the processes driving the changes are partly similar, implying that the contribution of GIS and WAIS to SLC are correlated. This correlated dependency is investigated by examining changes in the $90^{th}$ , $95^{th}$ and $97.5^{th}$ percentiles if dynamical ice sheet mass loss of GIS is fully correlated with climate driven changes in SLC. Dynamical ice sheet mass loss of WAIS to SLC is subsequently assumed to be correlated with 70 % this combined probability distribution (Appendix 2 Supplementary Methods). Mass loss of EAIS is still considered to be indepen-
dent of the other components. A result of the correlation is that the combined distribution is less wide, with low- and high-end percentiles that are closer to the mean. For the asymmetric distribution the high-end percentiles of SLC will therefore be lower compared to simulations where the SLC-components are merged independently (Fig. A3). A correlation between the different components that contribute to SLC would therefore result in a smaller range of sea levels. A better physical understanding if and to what extend dynamical changes are coupled to to climate changes is therefore important.

### 3.4   Impact of different input distributions

To analyse the impact of different PDFs of dynamical ice mass loss on regional SLC, we computed the 90, 95, $97^{th}$ percentiles for projections that include probability distributions for EAIS and WAIS following Ritz et al. (2015). The GIS contribution are based on IPCC-AR5, the probability distributions are combined with the regional fields of Slangen et al. (2014). In these simulations SLC projections increase non-linear for the analysed percentile, e.g the increase is approximately the same between
$90^{th}$-$95^{th}$ and $95^{th}$-$97.5^{th}$ (Fig. 7a-c). A difference of the regional Ritz-projections compared to the regional IPCC-projections is that mainly in the Northern hemisphere the SLC values shift towards higher values, up to 0.2 m (Fig. 7d-f). This is related to the fingerprints of EAIS and WAIS, by mass loss in the far field of the ice sheet. The range of projected SLC is different in the regional Ritz-projections compared to the regional VW15-projections (Fig. A4). This is not only because the contribution





of GIS is based on IPCC-AR5, but also because the input distributions of EAIS and WAIS have a less heavy tail towards high SLC values (Fig. 1e-h). This analysis shows the importance of accurate probability distributions for dynamical ice sheet mass loss for higher percentiles of regional SLC which based on ice dynamical models.

## 4  Discussion

The asymmetric probability distributions for the contribution of ice sheet mass loss in this research are based on two projections of dynamical ice sheet mass loss. One of these studies (VW15) is based on an expert judgment data set. This approach has a number of limitations. Firstly, the interpretation of the expert data can largely influence the shape of the tail of the probability distribution (De Vries and Van de Wal, 2015). Secondly, according to Gregory et al. (2014) some surveyed experts to the expert analysis by Horton et al. (2014) suggested SLC projections outside a range of physically plausible scenarios. Furthermore, they

argue that an expert judgement analysis is an opaque way of data gathering and that especially outliers to high values cannot be verified. The study by Ritz et al. (2015) established a physically based probability distribution of dynamical ice sheet mass loss of the Antarctic ice sheet. Indications of a the possibility of a collapse of parts of ice sheets (Little et al., 2013), show that the contribution of the AIS to SLC is highly uncertain. For the GIS a physically based probability distribution of dynamical ice sheet mass loss is not available, it is not expected that the probability distribution of the GIS has a large tail towards high SLC

values, it would be good to replace expert judgement analysis with physically based probability distributions. The data sets for dynamical ice sheet mass loss to SLC available at this moment ((DeConto and Pollard, 2016) and Fig. 1) show a large range of values. In this study we show that when these global projections are used for regional SLC projections, the differences in higher percentiles will be amplified.

In Section 3.3 we analyse the impact of a correlation between climate induced changes to SLC and ice dynamical con-

tributions to SLC. We show that a correlation between different contributions to SLC impact SLC projections. The ratio of correlation is based on an expert judgement analysis (Bamber and Aspinall, 2013). Future research should determine physically based correlation factors. In general high correlations reduce the width of the combined PDFs, so a correlation between components that contribute to SLC may reduce the high percentiles.

In coastal safety assessment higher percentiles are often used to calculate return-frequency based extremes. The uncertainty

bands of these extreme events are often used to project if a specific event is changing significantly under a future climate. The projections of high-end uncertainties also have an uncertainty (De Vries and Van de Wal, 2015, their Figure 3 and 4), including this in future studies would make it possible to determine the bandwidth of the tail of the cumulative density function for SLC projections and analyse the significance of extreme SLC.

The method presented here could also be used to analyse the effect of (asymmetrical) uncertainties in other components that

contribute to SLC such as thermal expansion (Sriver et al., 2012), changes in ocean currents and temperature (Sallenger Jr. et al., 2012; Yin and Goddard, 2013), or non-climatological local effects (Santamaría-Gómez et al., 2014).





## 5 Conclusions

Until recently, SLC studies focused on projections with symmetric uncertainty ranges. Here, we have shown that the tail towards high values of SLC of the probability distribution of dynamical ice sheet mass loss highly influences the $90^{th}$, $95^{th}$ and $97.5^{th}$ percentiles of regional SLC. This shift of higher percentiles has large regional variability due to local differences in the contribution to SLC from dynamical ice sheet mass loss, related to the distance to the ice sheets of Greenland, East and West-Antarctica. Asymmetric distributions of dynamical ice sheet mass loss can affect the median of SLC projections, with a global-average shift in median of 0.18m for the simulations with the asymmetric distributions of dynamical ice sheet mass loss by De Vries and Van de Wal (2015) (VW15) compared to the symmetric distributions based on IPCC-AR5. This can be related to a higher input median of the dynamical ice sheet mass loss components and to the skewed tail of the probability distribution towards higher values. The shift of the higher percentiles is even more pronounced compared to the shift in median is the contribution of ice sheet dynamics to SLC is assumed to be asymmetrically distributed towards higher values. For the $97.5^{th}$ percentile the shift can be up to 0.54m and over 0.3 m if the local shift in median is taken into account between the asymmetric VW15 and the symmetric IPCC-based distributions.

If dynamical ice sheet mass loss of the Greenland and West-Antarctica is (partly) correlated to climate induced changes in sea level rise, the increase of SLC projections is slightly lower. However, even with this correlation, the $97.5^{th}$ percentile is locally exceeding 1.6m. The $90^{th}$, $95^{th}$ and $97.5^{th}$ percentiles of regional SLC are strongly effected by the analysed asymmetric probability distribution. The difference between the asymmetric input probability distributions of dynamical ice mass loss of De Vries and Van de Wal (2015) and Ritz et al. (2015) show that the high-end percentiles can differ up to 0.5m depending on the applied PDF for dynamical ice sheet mass loss. Hence, we conclude that the uncertainty in ice sheet dynamics dominates the uncertainty in the local high-end percentiles of SLC projections. This is highly relevant for flooding safety, since with a higher mean sea level critical thresholds are exceeded more frequently under extreme events, such as storm surges. Focussing on the median regional sea level change with a symmetric uncertainty range therefore underestimates future climate change related flooding risks.



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



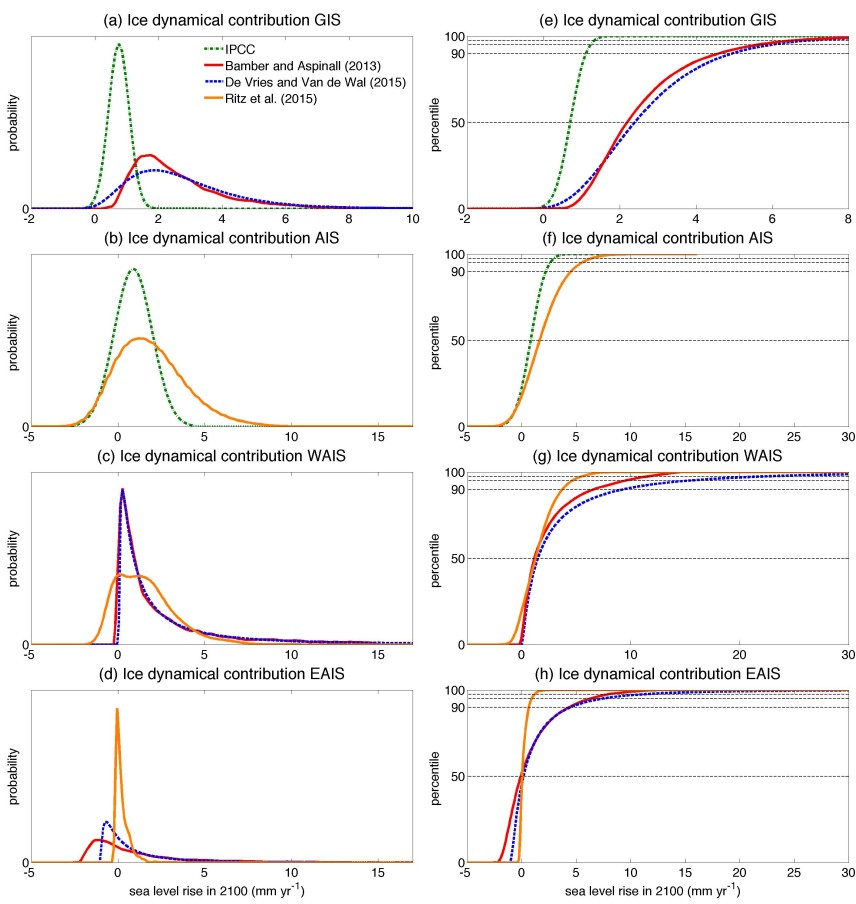

**Figure 1.** Probability density functions (PDF) (left column) and cumulative density functions (CDF) (right column) for dynamical ice sheet mass loss of Greenland ice sheet (GIS), Antarctic ice sheet (AIS), West-Antarctic ice sheet (WAIS) and East-Antarctic ice sheet (EAIS). IPCC-AR5 does not have separate distributions for WAIS and EAIS. In the CDFs the dotted lines indicate the $90^{th}$, $95^{th}$ and $97.5^{th}$ percentiles.




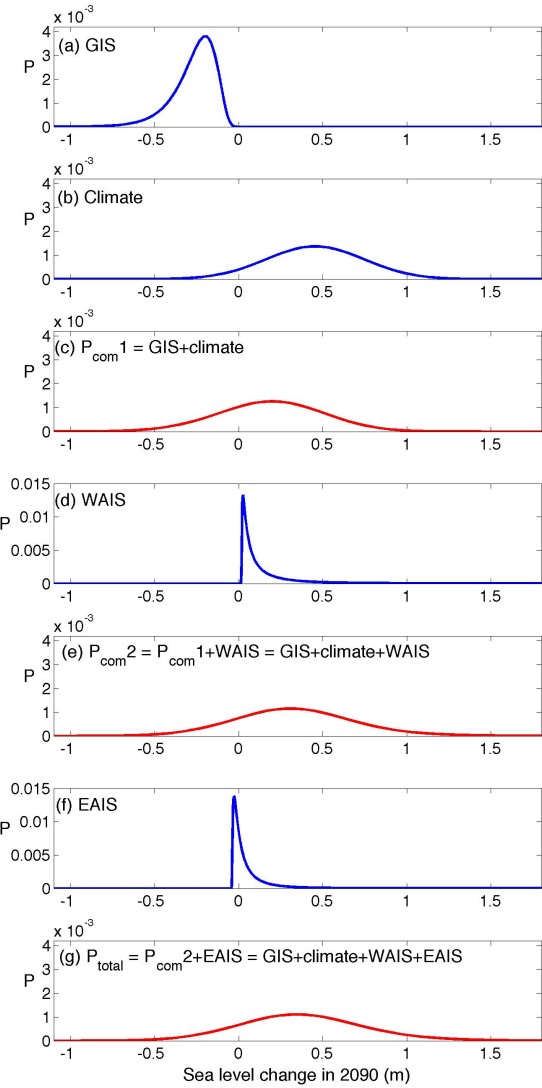

**Figure 2.** Example of the merging of several probability density functions (PDF), here depicted for Denmark Strait (3a). The input PDF of dynamical ice sheet mass loss of the Greenland ice sheet (GIS) and climate forcing are merged following Eq. 1, to calculate $P_{com}1$. This combined PDF is subsequently combined with a PDF of the ice dynamical contribution of the West-Antarctic ice sheet (WAIS) to SLC to $P_{com}2$, finally the ice dynamical contribution of the Eest-Antarctic ice sheet (EAIS) is added to construct $P_{total}$. The input PDFs (blue lines panels, a, b, d and f) vary regional, as depicted in Fig. A1 for 3 locations.




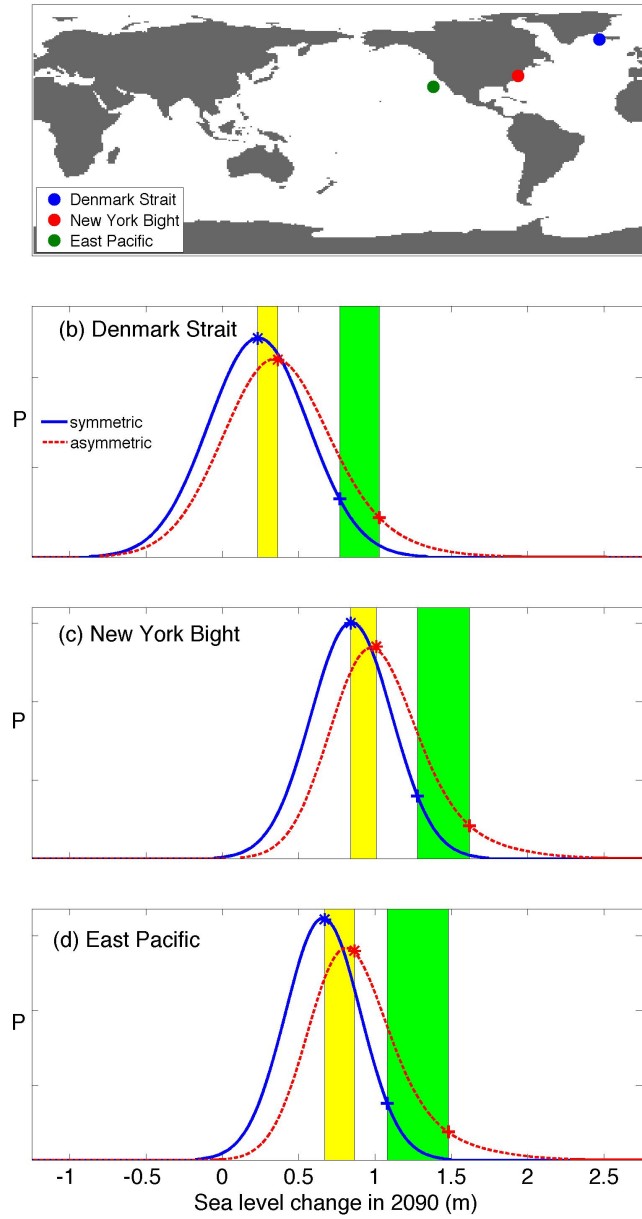

**Figure 3.** Total combined probability density of SLC by 2090, for 3 locations marked on panel (a): b. Denmark Strait, c. New York Bight and d. East Pacific. Regional projections by Slangen et al. (2014) following RCP8.5 combined with (blue, solid lines) the symmetrical IPCC-based distributions of ice sheet mass loss and (red, dotted lines) asymmetrical VW15-distribution of ice sheet mass loss. The stars indicate the median and the plus sign indicate the $95^{th}$ percentile. The difference in median is indicated in yellow and the difference in the $95^{th}$ percentile in green. Note that the increase of the $95^{th}$ percentile is larger than the increase in median. The input distributions for each location are depicted in Figure A1




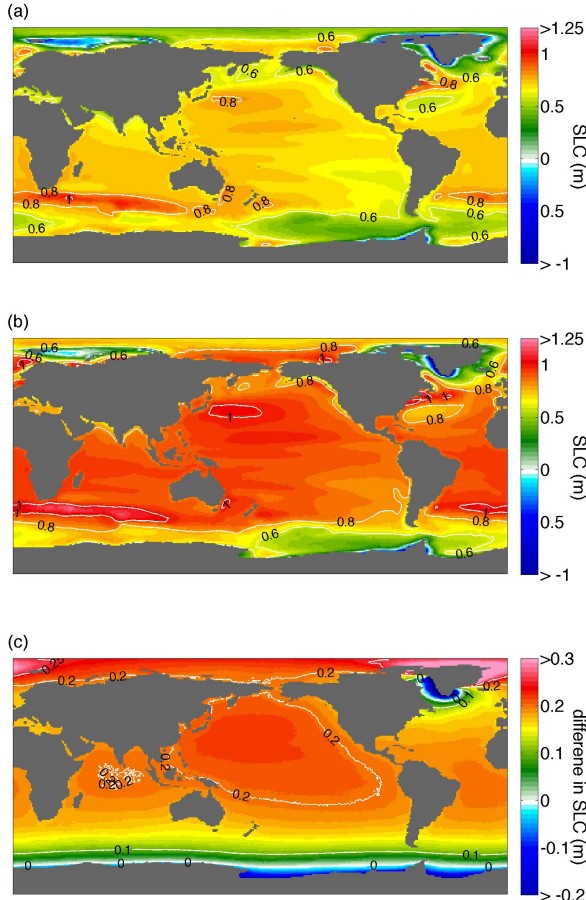

**Figure 4.** Median sea level rise projections by 2090 regional projections of Slangen et al. (2014) following climate scenario RCP8.5 combined with the contributions to sea level rise due to dynamical ice sheet mass loss: (a) symmetrically distributed based on Church et al. (2013) and (b) asymmetrically VW15 distributed (De Vries and Van de Wal, 2015). c. The difference in median (panel b − panel a). The area-averaged global median depicted in pannel a is 0.68m and 0.86m in pannel b. The area-averaged global median difference between pannel (b) and (a) is 0.18m.





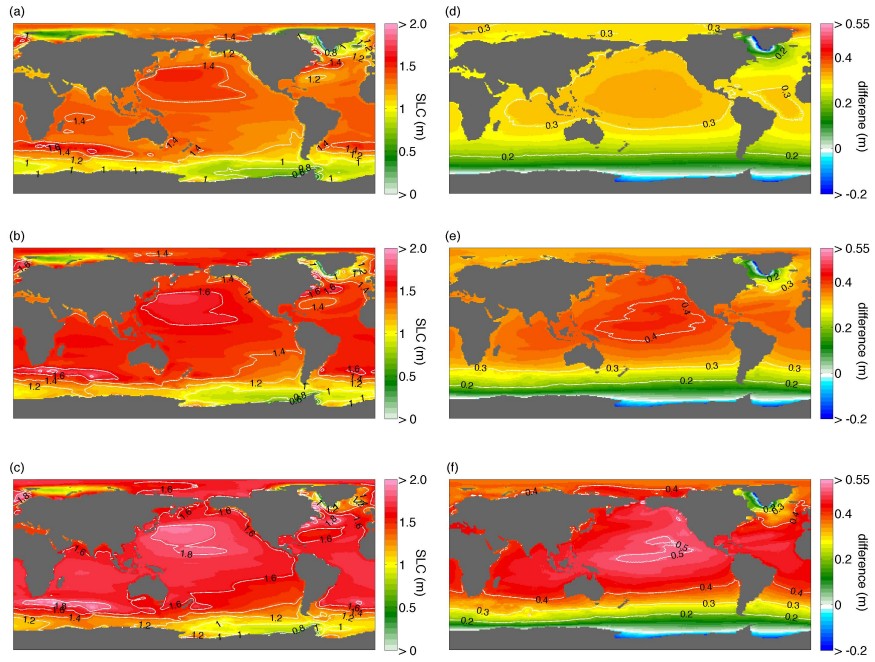

**Figure 5.** SLC projections for high-end percentiles and the change therein by 2090. Left column: a. $90^{th}$ , b. $95^{th}$ and c. $97.5^{th}$ percentile for the asymmetric VW15-distribution of dynamical ice sheet mass loss combined with regional SLC projections following RCP8.5 (Slangen et al., 2014). Right column: difference between d. $90^{th}$ , e. $95^{th}$ and f. $97.5^{th}$ percentile for asymmetric VW15 and symmetric IPCC-based contribution, both combined with regional SLC projections following RCP8.5 (Slangen et al., 2014). Percentiles for the symmetric IPCC-based probability distribution are depicted in Figure A2.

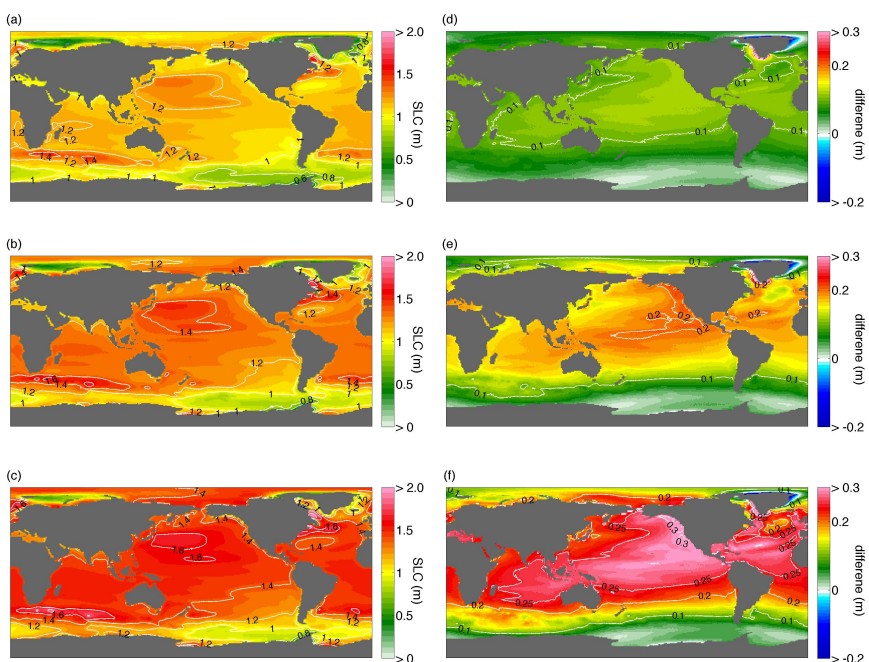

**Figure 6.** SLC projections by 2090 for high-end percentiles corrected with shift in local median. Left column: a. $90^{th}$ , b. $95^{th}$ and c. $97.5^{th}$ percentile for the asymmetric VW15 distribution of ice sheet mass loss combined with regional SLC projections following RCP8.5 (Slangen et al., 2014), corrected with the local shift in median depicted in Fig. 2c. Right column: difference between the asymmetric VW15 (corrected with the difference in local median) and symmetric IPCC-based (both combined with regional SLC projections following RCP8.5 (Slangen et al., 2014)) for the d. $90^{th}$ , e. $95^{th}$ and f. $97.5^{th}$ percentile.


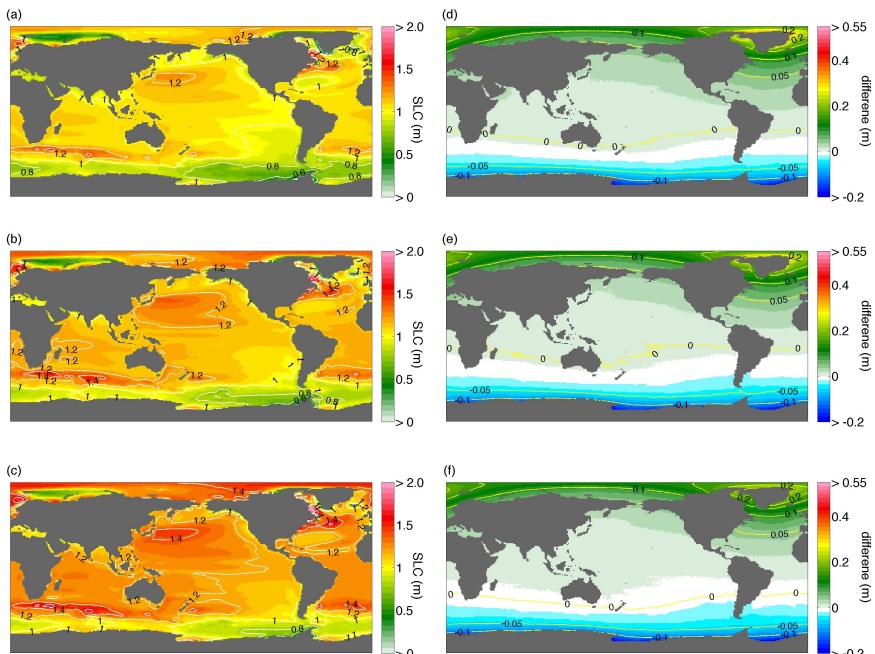

**Figure 7.** Left column: a. $90^{th}$ , b. $95^{th}$ and c. $97.5^{th}$ percentile for the asymmetric Ritz-distribution of dynamical ice sheet mass loss of East and West-Antarctica ice sheets (EAIS and WAIS) combined with symmetric IPCC-based contributions of dynamical ice mass loss of the Greenland ice sheet (GIS) and regional SLC projections following RCP8.5 (Slangen et al., 2014). Right column: difference between d. $90^{th}$ , e. $95^{th}$ and f. $97.5^{th}$ percentile for asymmetric Ritz and symmetric IPCC-based ice dynamical contribution of EAIS and WAIS, both combined with symmetric IPCC-based contributions of dynamical ice mass loss of the Greenland ice sheet (GIS) and regional SLC projections following RCP8.5 (Slangen et al., 2014). Percentiles for the symmetric IPCC-AR5 probability distribution are depicted in Figure SA2



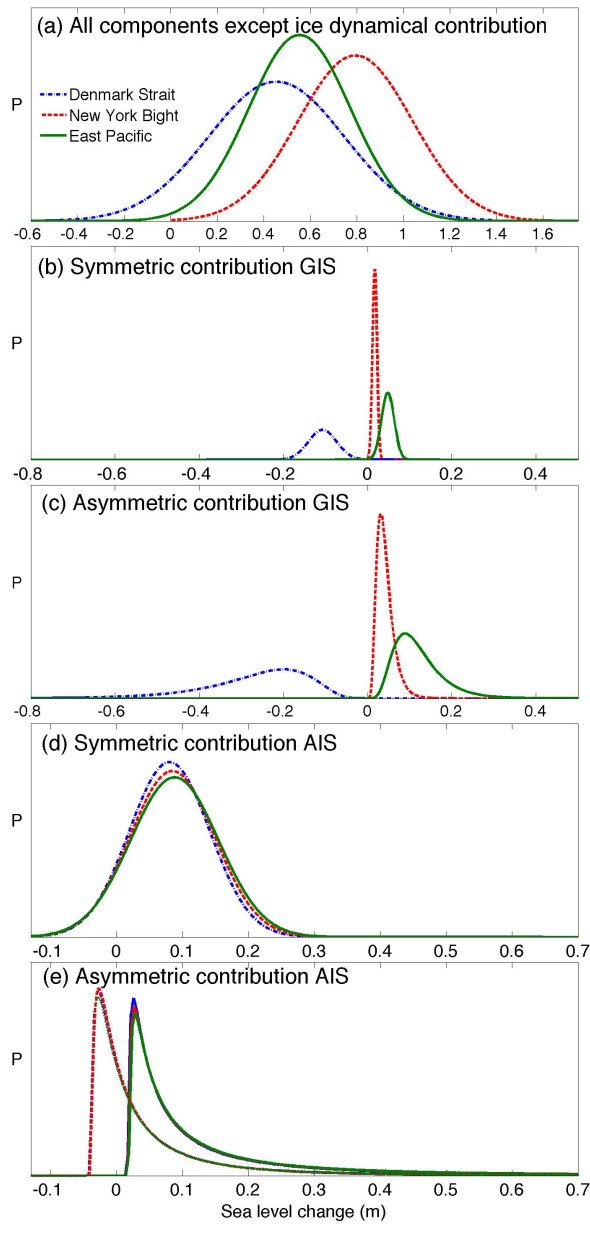

**Figure A1.** Contributions of dynamical ice sheets mass loss to sea level change at 3 locations (depicted in Figure 3a). a. All regional changes in sea level rise (SLC), except the dynamical ice sheet mass loss of Greenland and Antarctica, following RCP8.5 Slangen et al. (2014), The dynamical ice sheet contribution to SLC of the Greenland ice sheet based on: b. the symmetric IPCC-AR5 probability distribution (Church et al., 2013) , c. the VW15 asymmetric probability distribution (De Vries and Van de Wal, 2015). The dynamical ice sheet contribution to SLC of the Antarctic ice sheet based on: d. the symmetric IPCC-AR5 probability distribution (Church et al., 2013), e. the VW15 asymmetric probability distribution (De Vries and Van de Wal, 2015) with a distinction for East Antarctica only and West Antarctica only. Note the large tail to the left side with negative SLC-values for the GIS contribution (b,c) for the Denmark Strait located close to the GIS and the large tail towards positive SLC values (e) for the asymmetric EAIS and WAIS contributions.




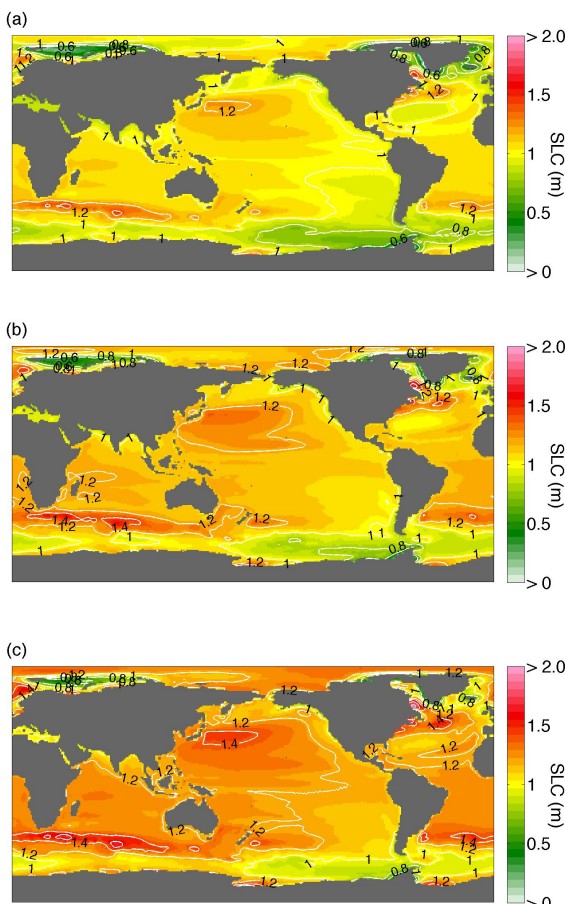

**Figure A2.** SLC projections for high-end percentiles by 2090 with symmetric IPCC-based distribution of dynamical ice sheet mass loss combined with the regional SLC projection of Slangen et al. (2014) following climate scenario RCP8.5, a. $90^{th}$, b, $95^{th}$ and c, $97.5^{th}$ percentiles.




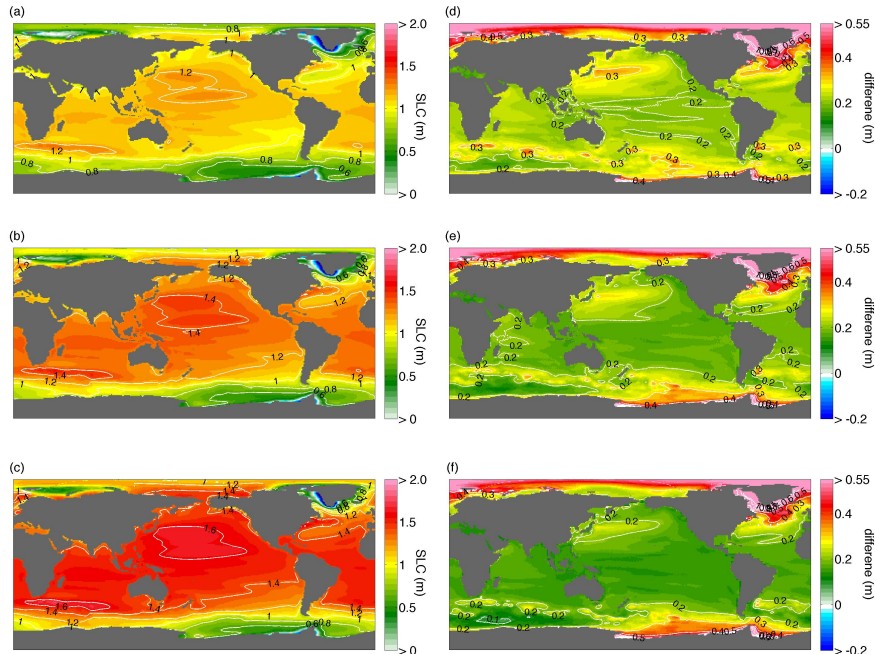

**Figure A3.** SLC projections for high-end percentiles and the change therein by 2090 if asymmetric VW15 dynamical ice sheet mass loss is partly correlated with climate change induced SLC changes. Left column: a. $90^{th}$, b. $95^{th}$ and c. $97.5^{th}$ percentile for the asymmetric VW15 distribution of dynamical ice sheet mass loss combined with regional SLC projections following RCP8.5 (Slangen et al., 2014). It is assumed that the dynamical ice sheet mass loss of Greenland ice sheet is fully correlated with all other regional changes in SLC; Dynamical ice sheet mass loss of West-Antarctic ice sheet is assumed to be by 70% correlated with this combined probability distribution. The dynamical ice mass loss contribution of East-Antarctic ice sheet is assumed to be fully uncorrelated with the other components. Right column: difference between the uncorrelated - partly correlated simulations (both combined with regional SLC projections following RCP8.5 (Slangen et al., 2014) and VW15 distributions) for the d. $90^{th}$, e. $95^{th}$ and f. $97.5^{th}$ percentile.





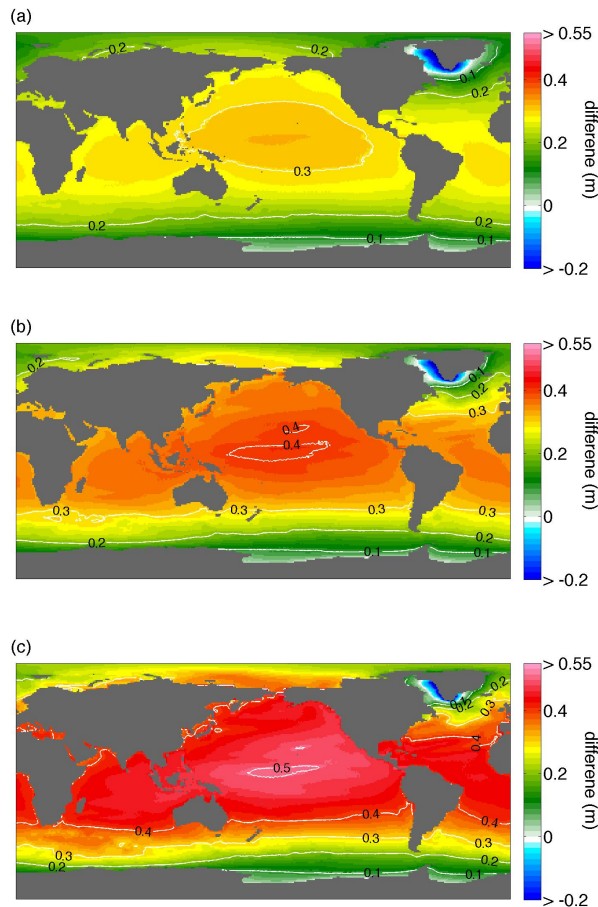

**Figure A4.** Difference between the a. $90^{th}$, b. $95^{th}$ and c. $97.5^{th}$ percentile for the asymmetric VW15 simulations (dynamical ice sheet mass loss of the Greenland, East and West Antarctic ice sheet (GIS, EAIS and WAIS) according De Vries and Van de Wal (2015)) and the asymmetric Ritz simulations (dynamical ice sheet mass loss of EAIS and WAIS according Ritz et al. (2015), GIS following the symmetric IPCC-AR5 distribution) both combined with regional SLC projections of Slangen et al. (2014) following climate scenario RCP8.5.





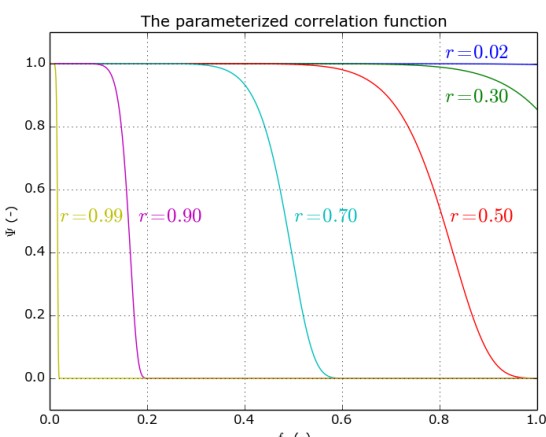

**Figure A5.** $\Psi(f_d(x_1, x_2))$. The correlation $r$ is a parameter in $\Psi$ which varies from 0 to 1. Here $\Psi$ is shown as function of $f_d$ for several $r$-values, as indicated by the labels in the figure





## 2   Supplementary Methods

In order to analyse the effect of dependent components that contribute to SLC, Eq. 1 has to be generalised for combining correlated components. Therefore a correlation function is constructed, which uses fractional intervals instead of $x$ values. For each distribution the interval between the left and the right boundary is scaled from 0 to 1. The fractional interval $f_1$ for $P_1$ is
defined by:

$$f_1 = \frac{x_1 - x_a}{x_b - x_a} \tag{1}$$

and the fractional interval $f_2$ for $P_2$ is defined by:

$$f_2 = \frac{x_2 - x_c}{x_d - x_c} \tag{2}$$

If the distributions of the two combined components are fully correlated then only the combination of $P_1$ ($f_1$) and $P_2$($f_2{=}f_1$)
contribute to the combined distribution. When $P_1$ and $P_2$ are assumed to be independent all combinations of $f_1$ and $f_2$ yield contributions to the combined distribution.

Therefore the fractional difference $f_d$ is defined as:

$$f_d = |f_1(x_1) - f_2(x_2)| \tag{3}$$

which indicates how far the fractional intervals of $x_1$ and $x_2$ are separated: e.g. if $f_d = 0$ then $x_1$ and $x_2$ are at the same
fractional interval.

Subsequently a correlation function $\Psi(x_1, x_2)$ is constructed which is a function of $f_d$ at the 0–1 interval. This function is parameterised by the correlation $r$: High $r$-values generate contributions to the combined distributions if $x_1$ and $x_2$ are close to each other, while $r = 0$ indicates that the distributions are independent. A function which satisfies this criteria and which is used in the analysis is:

$$\Psi(x_1, x_2) = e^{-\left(\frac{0.6 f_d(x_1, x_2)}{1 - |r|}\right)^{12}} \tag{4}$$

where the numbers 0.6 and 12 control the correct behaviour of $\Psi$, but their precise values are slightly arbitrary. Examples of three different $r$ values are shown in Fig. SA5. Over the interval $r = 0 - 1$ Eq. 4 is defined as:

$$\Psi(x_1, x_2) = \begin{cases} 1 & \text{independent distributions} \\ e^{-\left(\frac{0.6 f_d(x_1, x_2)}{1 - |r|}\right)^{12}} & \text{for} \\ 0 & a \leq x_1 \leq b \,\&\, c \leq x_2 \leq d \\ & \text{else} \end{cases} \tag{5}$$





The generalised form of Eq. 1 for combining two correlated SLC probability distributions $P_1$ and $P_2$ becomes:

$$P(x) = \sum_{m=c}^{d} P_1(x - m\Delta x) P_2(m\Delta x) \Psi(x - m\Delta x, m\Delta x) \tag{6}$$

which reduces to Eq. 1 in case that $P_1$ and $P_2$ are independent.

The numerical form of Eq. 6 adopted in the study is:

$$5 \quad P(i - i_0) = \sum_{m=c}^{d} P_1(i - m) P_2(m) \Psi(i - m, m) \tag{7}$$

where $i_0$ is the counter along the $x$-axis at $x = 0$, so $c$ and $d$ could be taken relative to $i_0$. Note that the correlation function $\Psi$ depends on the interval boundaries $x_a$, $x_b$, $x_c$ and $x_d$ as well. Other correlation functions which satisfy the global behaviour in Fig. SA5 for such $r$-values may be used, but do not yield qualitatively different results for combining correlated probability distributions.

10 *Author contributions.* RvdW conceived the study and designed the method with TR. TR developed the SEAWISE software. RdW carried out the analysis and wrote the manuscript under guidance of RvdW. RdW and TR wrote the Method Section. AS, HdV and TE prepared the data for the different contributions to SLC. All authors contributed to editing the manuscript.

*Competing interests.* The authors declare that they have no conflict of interest.

*Acknowledgements.* RdW and AS acknowledge the ALW-NPP program of NWO. TR is funded by Netherlands Earth System Science Center.