# Peer review of "Impact of asymmetric uncertainties in ice sheet dynamics on"

_Natural Hazards and Earth System Sciences, 2017_

## Short Comment (SC1) · 15 May 2017

First of all, I enjoyed reading the paper. The tails of the probability density functions (pdf's) are important and needed in the design of coastal defense infrastructure, and as such the paper is policy relevant.

I was wondering whether possible long term changes (intensification) in the terrestrial waster cycle (e.g. change in surface waters and soil moisture) are accounted for? The throughput of the water cycle is large so their might be a nett effect on sea level, and it might be widen/shift the pdf's of the type shown in fig 2 because of the added uncertainties. I'm asking because from GRACE we found some indications that water storage over the last decade has a negative contribution to sea level (see refs below).

References: Reager, J. T., Gardner, A. S., Famiglietti, J. S., Wiese, D. N., Eicker, A.,

& Lo, M.-H. (2016). A decade of sea level rise slowed by climate-driven hydrology. Science, 351(6274), 699–703. https://doi.org/10.1126/science.aad8386

Rietbroek, R., Brunnabend, S.-E., Kusche, J., Schröter, J., & Dahle, C. (2016). Revisiting the Contemporary Sea Level Budget on Global and Regional Scales. Proceedings of the National Academy of Sciences, 201519132. https://doi.org/10.1073/pnas.1519132113

Llovel, W., Becker, M., Cazenave, A., Crétaux, J.-F., & Ramillien, G. (2010). Global land water storage change from GRACE over 2002–2009; Inference on sea level. Comptes Rendus Geosciences, 342(3), 179–188. https://doi.org/10.1016/j.crte.2009.12.004

Riva, R. E. M., Bamber, J. L., Lavallée, D. A., & Wouters, B. (2010). Sea-level fingerprint of continental water and ice mass change from GRACE. Geophysical Research Letters, 37(19), L19605. https://doi.org/10.1029/2010GL044770

---

## Referee Comment (RC1) · Anonymous Referee #1 · 14 Jun 2017

Reviewer comments of:

"Impact on asymmetric uncertainties in ice sheet dynamics on regional sea level projections" by Renske de Winter et al.

recommendation:

minor revision

general comments:

The paper copes with the problem of propagating non-symmetrical sea level uncertainty distributions from ice sheet and glacier melting into the combined sea level rise. Since the authors use spatially varying self attraction and loading sea level change patterns, the estimates are furthermore varying in space. Compared to the case where symmetric distributions were used, the authors find mostly positive increases in the median of sea level change, and even larger increases in the higher percentiles of these projections. Furthermore, the authors study the effect of correlated contributors and alternative probability density functions.

I found the paper easy to read, with a clear message, and it may possibly be suitable for policymakers. I would therefore recommend the paper for publication. There are however a few minor issues which, when addressed, would improve the paper in my opinion.

* Explain the link between the combination of theoretical pdf's and the discretized formula's as provided in the paper. For example, eq 1 is a discretized convolution over the domain (-infty,infty) which comes from summing 2 contributions each with a different pdf. By briefly explaining the theoretical origin of eq. 1(and 6), one could make the paper more accessible to readers not so familiar with probabilistic theory.

* Uses of high percentile SLC estimates for coastal defense. This got me admittedly somewhat confused. As far as I understood, and I could be wrong, coastal defense infrastructure is commonly determined from high percentiles values of storm surge from models subjected to prescribed sea level rise, and not so much from the direct high percentile of this sea level rise itself. So my request would be to describe more clearly how these high percentile SLC values enter safety standards, rather than simply saying that they are used to define safety standards.

* Motivate choice of picking out locations Denmark Strait, New York and East Pacific. Why did the authors choose these locations? I can also imagine that locations in the West Pacific and Indian Ocean where large mega-cities exists will be highly relevant, not to mention that they are in the far field.

minor remarks:

page 2 l19: "is under debate": is it possible to add a reference here to a paper discussing this debate?

p2 l30 " An asymmetric probability density function for the Greenland .. can also no be included" Why is this? due to instability in the marine terminating glaciers? l30 also no -> also not

p3 l12 use distribution -> use a distribution

p3 l13 of by Bamber -> from Bamber

p3 l33 are published -> have been published

p6 l10 from -1.9 m to +1.03 m -> from -1.09m close to the melting sources to +1.03m in the far field

p7 l3 -> Adopting an asymmetric -> As mentioned before, adopting ..

p7 l6 Explain *why* you corrected the change of the higher percentiles for the local median SLC

p7 l24 Maybe add: as it can potentially narrow down the uncertainty of SLC projections

p8 l17 that when -> that, when

p8 l20 SLC projections -> its projections

p8 l20-21 "The ratio .. expert judgment" Would it be fair to mention that an increase in temperature in the climate may partly explain such correlations?

fig 2 caption Eest -> East

---

## Referee Comment (RC2) · Anonymous Referee #2 · 17 Aug 2017

The paper presents impacts of an asymmetric probability distribution of ice sheet dynamics on regional sea level projections using mass loss distributions of ice sheets from three studies. The topic is relevant for adaptation decision making as not only estimates of sea level rise need to be taken into account but also the uncertainties of these assessments. The paper is clearly written. I read it with great interest. I recommend to accept the paper with some minor revisions: - From the paper it was not clear to me what is/are the reasons for assuming an asymmetric distribution (p2 lines 21-27). What are the physical processes that make this plausible? Ice cliff instability? What is causing the shift from median to asymmetric distributions (page 4 first line)? New assumptions? What are they? It is addressed in the discussion, but I would like to have read it in the introduction - Page 3 line 17, where you describe the objective of the pa-

per. Maybe change this into: ..by comparing the impacts of probability distributions of... - Line 16 is confusing. Reinterpretation and using data (of what?) from .. is vague. I would remove the sentence here and explain in method section. - A flowchart/diagram showing the data used and the calculations made could improve understanding the method and the contribution of this paper in comparison to other studies. For example like fig 1 in Kopp et al 2014 10.1002/2014EF000239 - Could you explain why the difference in higher percentiles will be amplified (page 9 line18) - Figure 1 is 2100 and the other figures for 2090, why? - Would be great if there could be a dynamic figure with maps, where you could click on and see a graph like figure 3 Other comments - Page 8 line 16 → remove one ( - References: make consistent: De Vries or de vries - Add reference to Le Bars et al 2017 where you give example of symmetric pdfs

---

## Author Comment (AC4) · 6 Sep 2017

[revised manuscript text omitted]

As mentioned before, the globally-average value increases by +0.18m if the ice dynamical contribution is asymmetric compared to symmetric (Fig. 5c). However, this difference is much smaller than the difference in the higher percentiles, with a shift of the globally-average $97.5^{th}$ percentile of +0.39m. In order to determine whether the increase in sea level of higher percentiles is related to generally higher values (higher median) or to the shape of the distribution, we corrected the change of the higher percentiles for the change in local median SLC for each location (see Fig. 7 where Figure 5c is subtracted from Figure 6a-c). Generally, the corrected higher percentiles of SLC are still much larger compared to the projections with symmetric IPCC-based ice dynamical contributions, with changes up to 0.3m (Figure 7d-f). For both analysis (corrected and uncorrected for changes in the median), the increase to higher SLC values becomes more pronounced for higher percentiles. This has large implications for the high-end uncertainties of future SLC projections, since with a higher mean sea level the allowance for extreme events become lower as critical thresholds are exceeded more frequently (Slangen et al., 2017).

**3.3 Correlation between ice sheet mass losses and steric SLC**

[revised manuscript text omitted]

---

## Author Response (AR1)

general comments:

The paper copes with the problem of propagating non-symmetrical sea level uncertainty distributions from ice sheet and glacier melting into the combined sea level rise. Since the authors use spatially varying self attraction and loading sea level change patterns, the estimates are furthermore varying in space. Compared to the case where symmetric distributions were used, the authors find mostly positive increases in the median of sea level change, and even larger increases in the higher percentiles of these projections. Furthermore, the authors study the effect of correlated contributors and alternative probability density functions. I found the paper easy to read, with a clear message, and it may possibly be suitable for policymakers. I would therefore recommend the paper for publication. There are however a few minor issues which, when addressed, would improve the paper in my opinion.

1. Explain the link between the combination of theoretical pdf's and the discretized formula's as provided in the paper. For example, eq 1 is a discretized convolution over the domain (-infty,infty) which comes from summing 2 contributions each with a different pdf. By briefly explaining the theoretical origin of eq. 1(and 6), one could make the paper more accessible to readers not so familiar with probabilistic theory.
*The formula is introduced in more detail in the manuscript now by:*
"The composed distribution Pcom(x) consists for each x of all contributions of two independent distributions P1(x1) and P2(x2) for which the summed x-axis values x1 and x2 add to x. Each combination for which applies x = x1 +x2 yields a contribution of P1(x1) times P2(x2) to Pcom(x); i.e. summing over all relevant combinations x = x1+x2 determines Pcom(x) for a certain x:"

2. Uses of high percentile SLC estimates for coastal defense. This got me admittedly somewhat confused. As far as I understood, and I could be wrong, coastal defense infrastructure is commonly determined from high percentiles values of storm surge from models subjected to prescribed sea level rise, and not so much from the direct high percentile of this sea level rise itself. So my request would be to describe more clearly how these high percentile SLC values enter safety standards, rather than simply saying that they are used to define safety standards.
*The reviewer is correct that normally high percentiles are used as estimate for return levels. However, coastal decision making also requests information on the upper-boundary of possible future sea levels. To stress this following line is added to the manuscript:*
"Including high-end SLC projections is therefore the logical next step in coastal safety analysis, since coastal decision making also needs information on the upper-boundary of possible future sea level when assessing future extreme events (De Winter and Ruessink, 2017)"

3. Motivate choice of picking out locations Denmark Strait, New York and East Pacific. Why did the authors choose these locations? I can also imagine that locations in the West Pacific and Indian Ocean where large mega-cities exists will be highly relevant, not to mention that they are in the far field.

*These fields are chosen, because they provide a good insight in how different contributions contribute to the total PDF, hence it is not based on the relevance of the cities involved, but more as an introduction to understand the figures with a global coverage. Nevertheless, we expanded the number of location to have a wider geographical coverage of locations.*

minor remarks:

4. page 2 l19: "is under debate": is it possible to add a reference here to a paper discussing this debate?

*References to Vieli & Payne (2005) , Pattyn et al. (2012) and DeConto and Pollard (2015) are added.*

5. p2 l30 " An asymmetric probability density function for the Greenland .. can also no be included" Why is this? due to instability in the marine terminating glaciers? l30 also no -> also not

*This sentence is extended with* "not be excluded due to a rapid decay of marine terminating glaciers (Nick et al. 2013)*"*

6. p3 l12 use distribution -> use a distribution

*Thank you for the suggestion, this is changed.*

7. p3 l13 of by Bamber -> from Bamber

*Thank you for the suggestion, this is changed.*

8. p3 l33 are published -> have been published

*Thank you for the suggestion, this is changed*

9. p6 l10 from -1.9 m to +1.03 m -> from -1.09m close to the melting sources to +1.03m in the far field

*Thank you for the suggestion, this is changed to "from -1.90 m close to the location of mass loss to +1.03m in the far field"*

10. p7 l3 -> Adopting an asymmetric -> As mentioned before, adopting ..

 *Thank you for the suggestion, the sentence is changed to:*

"As mentioned before, the globally-average value increases by +0.18m if the ice dynamical contribution is asymmetric compared to  symmetric."

11. p7 l6 Explain *why* you corrected the change of the higher percentiles for the local median SLC

*Thank you for this suggestion, we explained in more detail why we did this, by adding the following sentences:*

"In order to determine whether the increase in sea level of higher percentiles is related to generally higher values (higher median) or to the shape of the distribution, we corrected the…"

12. p7 l24 Maybe add: as it can potentially narrow down the uncertainty of SLC projections

*In the Discussion section we added that this might also be related to the shape of the input PDF:*

"Le Bars et al. (2017) concluded, based on symmetric contributions, that the combined PDFs becomes wider, if the contributions are assumed to be correlated, suggesting that the shape of the distribution is also important."

13. p8 l17 that when -> that, when
*Thank you for the suggestion, this is changed*

14. p8 l20 SLC projections -> its projections
*Thank you for the suggestion, this is changed*

15. p8 l20-21 "The ratio .. expert judgment" Would it be fair to mention that an increase in temperature in the climate may partly explain such correlations?
*The correlation investigated in this study is between the climate driven components that contribute to sea level rise (which are largely temperature driven) and mass loss due to ice dynamical processes. The correlation factor used in this study is based on an expert judgement analysis. Future research should determine if there is a correlation between those processes. This is mentioned in the manuscript.*

16. fig 2 caption Eest -> East
*Thank you, this is changed*

**Anonymous Referee #2**

The paper presents impacts of an asymmetric probability distribution of ice sheet dynamics on region al sea level projections using mass loss distributions of ice sheets from three studies. The topic is relevant for adaptation decision making as not only estimates of sea level rise need to be taken into account but also the uncertainties of these assessments. The paper is clearly written. I read it with great interest. I recommend to accept the paper with some minor revisions:

1. From the paper it was not clear to me what is/are the reasons for assuming an asymmetric distribution (p2 lines 21-27).What are the physical processes that make this plausible? Ice cliff instability? What is causing the shift from median to asymmetric distributions (page 4 first line)? New assumptions? What are they? It is addressed in the discussion, but I would like to have read it in the introduction

*The following sentence is added:*
"This is due to non-linear behaviour of the ice dynamics and ice shelve collapse, and the possibility for a threshold affecting the rate of decay of the ice sheet-shelf system."

*Regarding your question "What is causing the shift from median to asymmetric distributions", the new studies cited in this particular line contain two elements: 1. a shift in median ($50^{th}$ percentile) and 2. a different shape. The sentence has been changed to make this more explicit to:*
"These new studies differ in median (indicated by the 50th percentile in the right column of Fig. 1) and asymmetry (shape of the PDF)"

2. Page 3 line 17, where you describe the objective of the paper. Maybe change this into: ..by comparing the impacts of probability distributions of...

*Thank you for addressing the issue, the objective can indeed be better formulated. In the new manuscript this is changed to:*
"The main objective of this paper is to analyse the sensitivity of higher percentile of regional SLC projections to asymmetric probability distributions for dynamical ice sheet mass loss. This is done by comparing the impacts of the probability distributions of Church et al. (2013), De Vries and Van de Wal (2015) and Ritz et al. (2015) on high-end regional SLC projections."

3. Line 16 is confusing. Reinterpretation and using data (of what?) from .. is vague. I would remove the sentence here and explain in method section.

*In line with the reply on the previous comment (2), the sentence the reviewer is referring to can be removed. Since this information is now included in the new sentence. We agree with discussing the reinterpretation only in the method section.*

4. A flowchart/diagram showing the data used and the calculations made could improve understanding the method and the contribution of this paper in comparison to other studies. For example like fig 1 in Kopp et al 2014 10.1002/2014EF000239

*Thank you for this suggestion. We added a flowchart to provide more insight in the computations.*

5. Could you explain why the difference in higher percentiles will be amplified (page 9 line18)

*Calculating global average SLC to regional SLC-patterns results to an above global-average increase in sea level, in locations in the far field of an ice sheet. This sentence is rephrased: "…the differences in higher percentiles will be amplified for locations in the far field of an ice sheet."*

6. Figure 1 is 2100 and the other figures for 2090, why?

*Figure 1 is based on the data as presented in the cited studies. The Slangen et al. (2014) study covers 2080-2100. In order to match the contribution of dynamical ice sheet mass loss to SLC to these regional projections, the data is within the Seawise program converted to 2090, as explained in line xx*

7. Would be great if there could be a dynamic figure with maps, where you could click on and see a graph like figure 3

*We agree that this would be nice; we will keep it in mind for future projects. Within this study we extended the number of locations in Figure 4. We are open to provided more detailed data on request.*

Other comments

8. Page 8 line 16→remove one (

*Thank you for the suggestion, this is changed*

9. References: make consistent: De Vries or de vries

*Thank you for the suggestion, this is changed*

10. Add reference to Le Bars et al 2017 where you give example of symmetric pdfs

*done*

[revised manuscript text omitted]

In Section 3.3 we analyse the impact of a correlation between climate induced changes to SLC and ice dynamical contributions to SLC. We show that a correlation between different contributions to SLC impact SLC projections impacts the high-end percentiles of the projections to generally slightly lower values. The ratio of correlation is based on an expert judgement

analysis (Bamber and Aspinall, 2013). Le Bars et al. (2017) concluded, based on symmetric contributions, that the combined PDFs becomes wider, if the contributions are assumed to be correlated, suggesting that the shape of the distribution is also imported.

5    In coastal safety assessment higher percentiles are often used to calculate return-frequency based extremes. The uncertainty bands of these extreme events are often used to project if a specific event is changing significantly under a future climate. The projections of high-end uncertainties also have an uncertainty (De Vries and Van de Wal, 2015, their Figure 3 and 4). Including this in future studies would make it possible to determine the bandwidth of the tail of the cumulative density function for SLC projections and analyse the significance of extreme SLC.

10    The method presented here could also be used to analyse the effect of (asymmetrical) uncertainties in other components that contribute to SLC such as thermal expansion (Sriver et al., 2012), changes in ocean currents and temperature (Sallenger Jr. et al., 2012; Yin and Goddard, 2013), or non-climatological local effects (Santamaría-Gómez et al., 2014).

Furthermore, in the study of Slangen et al. (2017) SEAWISE is used to determine the impact asymmetric probability distributions on sea-level allowances.

15    **5    Conclusions**

Until recently, SLC studies focused on projections with symmetric uncertainty ranges. Here, we have shown that the tail towards high values of SLC of the probability distribution of dynamical ice sheet mass loss highly influences the $90^{th}$, $95^{th}$ and $97.5^{th}$ percentiles of regional SLC. This shift of higher percentiles has large regional variability due to local differences in the contribution to SLC from dynamical ice sheet mass loss, related to the distance to the ice sheets of Greenland, East and

[revised manuscript text omitted]

**Figure 3.** Example of the merging of several probability density functions (PDF), here depicted for Denmark Strait (4a). The input PDF of dynamical ice sheet mass loss of the Greenland ice sheet (GIS) and climate forcing are merged following Eq. 1, to calculate $P_{com}1$. This combined PDF is subsequently combined with a PDF of the ice dynamical contribution of the West-Antarctic ice sheet (WAIS) to SLC to $P_{com}2$, finally the ice dynamical contribution of the  East-Antarctic ice sheet (EAIS) is added to construct $P_{total}$. The input PDFs (blue lines panels, a, b, d and f) vary regional, as depicted in Fig. A1 for 3 locations.

[Figure]

**Figure 4.** Total combined probability density of SLC by 2090, for 3 locations marked on panel (a): b. Denmark Strait, c. New York Bight, d. East Pacific, e. North Sea, f. Mekong Delta Vietnam, g. Tasmin Sea (East of Australia) and h. south east of South-Africa. 
[revised manuscript text omitted]